# Semi-Supervised Image Stitching from Unstructured Camera Arrays

**DOI:** 10.3390/s23239481

**Published:** 2023-11-28

**Authors:** Erman Nghonda Tchinda, Maximillian Kealoha Panoff, Danielle Tchuinkou Kwadjo, Christophe Bobda

**Affiliations:** Department of Electrical and Computer Engineering, The Herbert Wertheim College of Engineering, University of Florida, Gainesville, FL 32611-6200, USA; m.panoff@ufl.edu (M.K.P.); dtchuinkoukwadjo@ufl.edu (D.T.K.); cbobda@ece.ufl.edu (C.B.)

**Keywords:** image stitching, self-supervised learning, image blending, unstructured camera arrays, scene representation

## Abstract

Image stitching involves combining multiple images of the same scene captured from different viewpoints into a single image with an expanded field of view. While this technique has various applications in computer vision, traditional methods rely on the successive stitching of image pairs taken from multiple cameras. While this approach is effective for organized camera arrays, it can pose challenges for unstructured ones, especially when handling scene overlaps. This paper presents a deep learning-based approach for stitching images from large unstructured camera sets covering complex scenes. Our method processes images concurrently by using the *SandFall* algorithm to transform data from multiple cameras into a reduced fixed array, thereby minimizing data loss. A customized convolutional neural network then processes these data to produce the final image. By stitching images simultaneously, our method avoids the potential cascading errors seen in sequential pairwise stitching while offering improved time efficiency. In addition, we detail an unsupervised training method for the network utilizing metrics from Generative Adversarial Networks supplemented with supervised learning. Our testing revealed that the proposed approach operates in roughly ∼1/7th the time of many traditional methods on both CPU and GPU platforms, achieving results consistent with established methods.

## 1. Introduction

Image stitching is an important technique in many computer vision applications. Image stitching aims to combine multiple images captured from different viewpoints into a single image with a wider field of view (FOV) that encompasses all contributing images. Image stitching is a well-studied topic with widespread applications [1,2,3,4,5], and has proven to be very useful in domains such as virtual reality, teleconferencing, sports broadcasting, and immersive technologies [6,7,8]. However, existing stitching methods, despite their broad adoption, do not scale well to systems with many unorganized cameras.

State-of-the-art techniques often use sequential pairwise image stitching to generate panoramic images from multiple cameras [8,9,10]. Sequential Pairwise is a multi-step process in which two images are stitched simultaneously. At every step, a new image is stitched with the composite image of all already stitched ones. This image stitching technique simplifies the problem of complex overlapping regions introduced by multi-cameras, making it easy to find the intersection of the overlapping regions for efficient processing. However, this sequential process has several issues regarding time complexity and error propagation. An error introduced during an early merge is likely to be maintained rather than corrected, leading to many issues in the final output. Additionally, by only examining two sub-images, current solutions often fail to properly merge the content in the resulting image, leading to broken objects and ghosting effects.

As an alternative, seam-based approaches provide a way to limit error propagation by differentiating the pixels that previously belonged to a seam; however, this approach introduces seam-related artifacts [3,11]. Usually, an additional step is required to remove these artifacts. Methods such as Poisson blending [4] are often the primary postprocessing choice to remove color artifacts; however, Poisson blending assumes that the boundary between the overlapping pairs of images (e.g., not the overlap of each image but the overlap between each pair of images) is well-defined. In a multi-camera system with unorganized camera arrays, this assumption does not stand. There may be many individual sets of images, complicating the optimization process used in Poisson blending. As an alternative to Poisson blending, multi-band blending bypasses complications stemming from large amounts of overlapping images; however, it requires many iterations to convey an acceptable outcome under the same conditions, raising the issue of time complexity.

It is common to find approaches in the literature that tackle each of these problems individually very well; unfortunately, these methods are rarely able to produce an artifact-free panorama stream in real time when used with more complex and heterogeneous setups involving a large number of cameras [12]. In some approaches, significant computational resources are required to remove visual artifacts. For example, seam-based methods require additional processing, such as Poisson image blending, to eliminate visual artifacts caused by camera exposure [11,13,14].

A few recent works have examined image stitching using deep learning [15,16,17,18]. However, all of these focused on pairwise stitching, wherein two inputs are stitched together and the output is stitched with the next input until all inputs have been used, similar to the traditional methods discussed above. This can lead to several problems, in particular error propagation, as a single error early in the process can cascade; with each iteration, pre-existing errors are built upon and expanded. In our work, we stitch several images together concurrently, which mitigates this potential shortfall and leads to higher efficiency when using a large number of cameras.

Deep learning is a powerful tool in computer vision, and recent advances in image generation, particularly through Generative Adversarial Networks (GANs), have inspired a large body of work. In a GAN, a generator creates an image, and a discriminator scores or rates it, with the ultimate goal being the creation of an image that can fool the discriminator [19]. Image stitching is similar in scope; the central idea is finding a method to create a new image from several smaller ones such that the resulting image maintains all the information, content, and structure from the constituent images. In [20], Shen et al. proposed using a GAN for image stitching; however, their solution has a few key weaknesses. Notably, their work focuses on pairwise stitching and requires a precomputed and entirely accurate binary mask to highlight the overlap between image pairs when generating an image, which adds additional computational overhead.

In this paper, we introduce a pioneering deep learning-based method for stitching images from unstructured camera arrays. Our approach, distinct from Ye et al. [21], ensures the preservation of image content in the final stitched output without the need for binary masks during image generation. This method excels in handling the challenges of diverse viewpoints and lighting conditions through a custom loss function and innovative semi-supervised learning with a pre-trained perception model, significantly reducing the reliance on labeled data.

We present SandFall, an innovative algorithm designed for the efficient representation of image data. SandFall enables the processing of an unlimited number of camera inputs by compressing images into a fixed volume size, thereby minimizing data loss and addressing one of the key challenges in large-scale image stitching.

The primary contributions of our paper are:The development of SandFall, a novel algorithm for data representation that supports an unlimited number of camera inputs while ensuring fixed-size image data.A deep learning-based image stitching network capable of processing an unlimited number of input images in a single forward pass, employing semi-supervised learning to optimize training with limited labeled data.Adaptations of GAN performance metrics specifically tailored for assessing the quality of image stitching, providing a robust framework for evaluation.Comprehensive quantitative and qualitative evaluations of our approach, demonstrating its effectiveness against popular image stitching algorithms using both standard and custom datasets.

The remainder of the paper is structured as follows: Section 2 covers necessary background knowledge and related work in image stitching. Section 3 details our proposed framework and the SandFall algorithm. In Section 4, we demonstrate our technique and benchmark it against state-of-the-art methods. Our findings are discussed in Section 5, and we conclude with reflections on our approach in Section 6. The complete source code for this project, encompassing all algorithms and methods discussed, is publicly available for reference and further research. It can be accessed on GitHub as of 22 November 2023 at https://github.com/smartsystemslab-uf/unsup_image_stitching.

## 2. Related Work

### 2.1. Image and Video Stitching

Image stitching aims to create seamless and natural photo-mosaics using multiple images from different sources. A comprehensive survey of traditional image stitching algorithms is provided in [5]. Recent studies have focused on structure deformation and its extension to video stitching [14,22]. These approaches assume a pairwise overlapping of cameras and use dynamic programming to search for the optimal seam, which is unsuitable for unstructured multi-camera systems. To handle unstructured camera arrays, Perrazi et al. [10] used a straightforward approach for camera alignment by applying pairwise homography directly on the input videos. While this makes the technique more flexible and straightforward for camera alignment initialization, it requires additional processing to handle lens distortion and exposure compensation.

Machine learning, particularly deep learning, is highly impactful in several domains, particularly computer vision. Convolutional Neural Networks (CNNs), a type of deep learning architecture, are extremely successful when applied to traditional computer vision problems such as image classification [23], object detection [24], image segmentation [25], and human pose detection [26].

Image stitching has received far less attention from deep learning experts than these subdomains, though this is not to say there has been no prior work at all. Song et al. used CNNs in [16,18], making use of weak supervision and expanding their network to work with images taken in a simulated outdoor environment, which can be more difficult as these images have more variation in exposure levels. In [15], Chilukuri et al. stitched two images together and leveraged auto-encoders [21] in addition to standard convolutional layers when constructing their network. Specifically, they encoded two input images into a shared space and then decoded the result into a single output image. Shen et al. proposed a method in [20] involving the use of a Generative Adversarial Network to stitch two images with an overlapping field of view together using a CNN. Their work heavily leveraged a mirror to finely tune the amount of overlap between the fields of view of the images and to create perfectly aligned images for use as ground truth. However, while their proposed network introduces low amounts of artifacts and is able to run in real-time, which are among the greatest challenges in image stitching, it requires a precomputed binary mask to highlight the overlap between the input images. Finally, in [17], Nie et al. proposed a method using deep learning to better solve the problem of rectangling in image stitching. Again, this requires a precomputed binary mask and only attempts to solve pairwise stitching.

### 2.2. Parallax-Tolerant Stitching

Many recent approaches have focused on addressing parallax-tolerant stitching. One variety of these approaches assumes that all images with the same projection center are parallax-tolerant. It is possible to manipulate images to meet this constraint by carefully rotating each camera in the scene [2,5]; however, many errors can be introduced through misalignment of these projection centers caused by objects moving during image acquisition or incorrect mitigation of lens distortion. These errors can be removed using Multi-Band Blending (MBB) [2], content-preserving warping [13], and seam selection [3]. MBB usually provides satisfactory results; however, several iterations may be required for the algorithm to converge, making it unsuitable for real-time video stitching. Therefore, in this paper we use content-preserving warping for camera alignment and a deep neural network to increase stitching throughput.

### 2.3. Gradient Domain Smoothing

The main challenge with the seam-based approach is finding a good compromise between the structure of an image and the visual perception of the seam. When the emphasis is on preserving the structure of the objects in the scene, the stitches appear with a more visible seam. Additional steps are often required to remove seam-related artifacts using the Poisson equation, as formulated by Perez et al. [4]. The Poisson equation is designed to blend the image based in the assumption that the boundary of the intersection area is well-defined. To the best of our knowledge, this equation has not been formulated for blending several images (more than two) simultaneously. One solution that is often provided in the literature involves formulating the problem in the frequency domain and then using a guidance vector to find and approximate the solution with FFT [27]. This reformulation is known as the Fourier implementation of Poisson Image Editing [27,28,29,30]. These algorithms effectively remove additional artifacts when the composite image is not multi-style. For instance, if one part of the scene is under shadow and the other part is under strong illumination, the resulting image tends to be either too bright or dark.

### 2.4. Supervised, Unsupervised, and Semi-Supervised Networks

As machine learning becomes increasingly popular, its limitations are becoming more apparent. One of the largest drawbacks of supervised learning (the most common machine and deep learning approach) is that it requires large datasets with accurate ground truth labels [31]. The algorithm learns by predicting an input and comparing its result with the known true result. Differences between these two sets are used to calculate a loss function, which the network then attempts to minimize. As it does this, the network’s predictions and the ground truth become more aligned and the model grows more accurate. Ideally, the loss eventually reaches a minimum value, resulting in the network’s outputs closely aligning with the ground truth.

It is not always possible to obtain a large dataset with accurate labels when training a network. One possible solution to this problem is unsupervised learning. In unsupervised learning, there are no known ground truth labels; instead, the characteristics of the data itself are used as labels. A classic example of this in deep learning is the use of autoencoders for noise reduction [31]. In these networks, the inputs are taken as labels. An encoder-decoder network might receive an image, perform convolutions to lower its resolution, deconvolve it to upscale it, and use the difference between the original image and output to calculate the loss. Following this approach, labels can be quickly and automatically generated rather than being found through human input or expensive computation.

Another popular solution, and the one that we use in this work, is known as semi-supervised learning [32], which combines the two prior approaches of supervised and unsupervised learning. Instead of needing to label an entire large dataset, a small subset of collected data is labeled. The network first trains on this data in the same manner as in supervised learning. Soon after, additional unlabeled data are added to the training set, which is then trained in an unsupervised manner. By priming the network with supervised data, it should have a better chance to converge to a low loss than one trained in an entirely unsupervised approach while requiring less labeled data than supervised learning.

### 2.5. Image Quality Assessment (IQA)

An important aspect of the training process is efficiently assessing the quality of the generated output in a way that correlates with human judgment. This task is challenging in the context of unstructured image stitching for two main reasons. First, the camera registration process that allows images to be aligned in a frame of reference prior to stitching consists of geometric transformations. These transformations often depend on the perspective of dominant objects in the picture. Homography, for example, seeks to favor dominant planar structures. Thus, the transformation matrices used for projecting images into a common warping space are obtained as a trade-off between the content of the image and the objective scene [2,5,10]. For this reason, it is difficult to design a ground truth dataset for an unstructured array of cameras without it being subject to the geometric-related error obtained during the registration process. Second, in unsupervised image stitching the goal is to compare the generated image to the warped images. Because of the geometric errors introduced during the alignment process, pixel-based metrics such as MSE, PNSR, and SSIM that assess the image quality through direct pixel-to-pixel comparison are not suitable for evaluating the quality of the generated image against the warped inputs. In addition, these metrics do not usually correlate with human judgment, as shown by [33,34,35]. For this reason, throughout this work we weight feature-based metrics most heavily when assessing the quality of images.

Recently, several metrics have been proposed to evaluate performance in GAN-based image generation models [33,34,35,36,37]. These metrics can be categorized as featured-based, as they evaluate the quality of images using high-level features from pretrained networks. As opposed to pixel-based metrics (SSIM, PNSR, MSE, etc.), which compute the similarity between two images by relying on pixel values directly, feature-based metrics correlate well with human perception [33]. The Fréchet Inception Distance (FID) [35] was created to evaluate the performance of GAN networks by measuring the Fréchet distance on the feature space between the real image dataset and the fake one. The FID has been widely adopted in the literature, along with other metrics such as the Inception Score (IS) and the LPIPS for IQA.

## 3. Method

Stitching pipelines typically begin with image registration or alignment, followed by optimal seam finding, image blending, and finally artifact removal. The alignment phase involves projecting the input image into a reference frame by estimating the mutual poses of the input images.

Our approach focuses on the post-image alignment phase and aims to generate a seamless output. We formulate the stitching problem as finding a parametric function G that maps a set of warped input images $X=\{w\left(X^{i}\right)\}$ into a seamless image target T [Equation 1]. For simplicity, we use $X^{i}$ as the warped image instead of $w(X^{i})$ in the following. In contrast to existing approaches that use the pairwise methodology, we have designed our model to assess the inputs simultaneously in order to generate the stitched output. Our system consists of two components: a data representation S and an image transformation network F; S takes a set of multiple images and generates an intermediate representation with a regular shape. The second component, F, uses the intermediate representation to generate the stitched image.
(1)$$G\left(X;\theta \right)=F\left(S\left(X\right);\theta \right)\to T$$

### 3.1. Data Representation

In light of the intricacy posed by the unstructured nature of our method, the primary goal of our data representation strategy is to facilitate fixed-size inputs for model training while enabling the network to simultaneously assess all pixels. This is especially crucial when dealing with multiple cameras, which introduces an added layer of complexity to representation and processing.

Historically, research has predominantly focused on pairwise image stitching. In this approach, some images, denoted as $I_{i}$ and $I_{j}$, are amalgamated to produce an intermediary image $I_{ij}$. This process is iterative. For example, $I_{ij}$ might be further combined with another image $I_{k}$ to yield $I_{ijk}$, and so on [9,10,12,38,39]. A significant vulnerability arises in this method: if an object’s representation is skewed during initial pairwise combinations due to seam placement, the error becomes embedded, potentially affecting subsequent stitching. In essence, when an error occurs, addressing it in subsequent iterations becomes exceedingly difficult. This cascading effect amplifies the risk of objects being depicted inaccurately or even omitted entirely in the final composite.

To address these limitations, we propose a straightforward yet effective method that enables stitching all images simultaneously for a direct scene representation. A fundamental way to achieve this is by creating a 3D volume wherein images are layered sequentially, as illustrated in Figure 1 (left). For RGB images, the total number of layers would be three times the camera count; however, the practicality of model training demands fixed data dimensions. To meet this requirement, we present the SandFall method, which adeptly converts any number of camera captures into a consistently sized tensor that is perfectly suited for smooth model integration.

#### 3.1.1. Weighted Mask Integration

For efficient operation of the SandFall algorithm, the inclusion of a weighted mask is essential. This mask assigns priority levels to pixels, indicating their importance during the “dropping” process. This mask assigns priority levels while actively modifying the pixel values based on the weighting scheme, ensuring that certain regions of the image are emphasized more than others during the warping and SandFall operation. This distinction is grounded in the idea that certain parts of an image are more crucial for preserving scene context and structure.

While evaluating functions to generate this mask, we found the sigmoid function to be particularly effective. This function creates a gradient of pixel priorities that originates from the center of the image. We considered the Gaussian function; however, its symmetric curve sometimes resulted in undesirable weightings at the image boundaries. In comparison, the sigmoid function offers better control over the transition between weights, making it more suitable for different image conditions.

An inverted sigmoid function was chosen in order to prioritize central pixels and reduce the weight of boundary pixels. The weight associated with the distance from the center of the image is defined as follows:(2)$$S^{\prime }\left(x,y\right)=1-\frac{1}{1+e^{-k(D\left(x,y\right)-d_{0})}}$$
where:D(x,y) is the Euclidean distance of a pixel (x,y) from the image center.*k* is a constant that adjusts the transition’s steepness.$d_{0}$ is a distance threshold that determines the transition’s starting point between high and low weights.

Figure 2 displays the variations of the inverted sigmoid function based on different *k* and $d_{0}$ values and the corresponding effects on the weighted mask.

#### 3.1.2. SandFall Algorithm

Upon obtaining the weighted mask using the aforementioned method, it is multiplied element-wise with the input image to produce a weighted image. This weighted image is subsequently warped to form the SandFall blob. This ensures that pixels closer to the image center have a more pronounced contribution during the warping process while those near boundaries exert reduced influence.

The foundational idea behind SandFall is that after warping the image into the shared volume, each image is used to constitute a layer of our 3D volume (as seen in Figure 1). Every pixel in the 3D volume is then “dropped” as if it were a grain of sand. It descends through the volume, replacing less significant or vacant pixels, until it reaches its rightful position or descends to the bottom-most row. This method ensures that the bottom layer retains the most comprehensive dataset, with data completeness diminishing for each subsequent upper layer.

During training, the layers within the 3D volume are occasionally shuffled in order to enhance the robustness of the model. This prevents the model from learning based on the order of the layers, ensuring that the algorithm’s efficacy is not compromised.

One notable advantage of SandFall is its scalability with increased camera counts. Because upper layers inherently contain less data, omitting many of these layers can reduce the volume’s size while only marginally impacting the overall result. For instance, in a context with N=100 cameras, which would usually necessitate a H×W×N×3 array, SandFall enables a reduction to a more manageable H×W×k×3 array (where k=5). This compact representation preserves significant scene data and aligns the input for deep learning architectures that mandate fixed input sizes. The marginal data loss, mainly when multiple camera perspectives overlap on a point, has an inconsequential bearing on the model’s performance. Users can adjust the *N* parameter to strike a balance between image space efficiency and the desired overlap.

In Figure 3, we illustrate the progressive construction of the stitched panorama using the SandFall method across four levels. Figure 3a depicts the base level where the majority of the complete data is present. Figure 3b–d shows progressively higher levels where fewer data points are included, demonstrating how our algorithm effectively manages overlapping and reduces the size of the dataset while maintaining the integrity of the scene. Notice how the images become more fragmented with each subsequent level, reflecting the selective inclusion of pixels based on their weighted importance.

### 3.2. Image Transformation Network

The pipeline for our stitching process can be seen in Figure 4. Our system consists of three main components: warping, reprocessing, and stitching. The system accepts as inputs a set of images $X^{i}$ consisting of *k* images taken at time *i*, each with three channels of H×W pixels. We then use the method described in [2] to project all images into a single four-dimensional (4D) plane of size H×W×k×3 (the fourth dimension is fixed to 3 in order to represent Red, Green, and Blue (RGB) for each image). We then utilize the SandFall Algorithm 1 to compress this to a smaller 3D plane of size H×W×N, where *N* denotes a user-selected limit, *n*, multiplied by 3 (again, once for each RGB channel). This limits the amount of data that the stitching algorithm has to process while defining an input matrix size that can represent an arbitrarily large number of cameras. At the same time, we create *k* masks for the resulting plane in order to identify all pixels in the 4D plane that belong to a single camera (i.e., the content of each warped image with only pixels where there are no intersections between any cameras) when training the network.
**Algorithm 1:** The SandFall Algorithm**Input:**   ● $X=\{I_{1},I_{2},\ldots ,I_{k}\}$: Set of RGB images of size H×W×3 where invalid pixels have value −1, ● $M=\{M_{1},M_{2},\ldots ,M_{k}\}$: Set of weighted masks, ● *n*: Max layers in output.**Output:** 
*V*: Matrix H×W×(3n).1:Initialize: $V_{t}\in R^{H\times W}=0,\;V\in R^{H\times W\times 3n}=0$.2:**for** 
(x,y)∈{1,…,W}×{1,…,H} 
**do**3:    **for** $(I_{k},M_{k})\in X\times M$ **do**4:        **if** $I_{k}\left[x,y\right]\neq -1$ **then**5:           $I_{k}\left[x,y\right]\leftarrow I_{k}\left[x,y\right]⊙M_{k}\left[x,y\right]$6:           $m\leftarrow V_{t}\left[x,y\right]$7:           $V\left[x,y,m\right]\leftarrow I_{k}\left[x,y\right]$8:           $V_{t}\left[x,y\right]\leftarrow m+1$9:        **end if**10:    **end for**11:**end for**12:**return** 
*V*

The network architecture for our stitching model consists of six inner convolutional blocks and one larger convolutional skip connection. Each convolutional block consists of the following layers (depicted in Figure 5). First is a 2D convolution with 64 1×1 kernels with a Rectified Linear Unit (ReLU) as the activation function, followed by a batch normalization layer, then a ReLU activation, another 2D 1×1 convolutional layer with 64 kernels and ReLU activation, and a final batch normalization layer. The output of this layer is summed with the input to the first convolutional layer of the block to form a residual connection, as described in [40]. Finally, the output of the first convolutional layer is summed with the output of the penultimate layer to form a skip connection. This can be seen in Figure 5. The output of the network is then compared to a reference through one of two methods described in Section 3.2.2.

#### 3.2.1. Overview

Let $X=\{X_{1},X_{2},...,X_{k}\}$ be a set of images captured simultaneously from a scene with $k_{c}$ cameras; $\omega \left(X\right)={\left\{\omega_{i}\left(X_{i}\right)\right\}}_{i=1}^{k}$ is the projection of each image $X_{i}$ into their respective warped space, e.g., an FOV where all pixels from each image can be seen, where $\omega \left(X\right)\in R^{H\times W\times 3*k}$. We compute the target image $y\in R^{H\times W\times 3}$ using slow but high-quality traditional image stitching methods and consider the resulting image as being stitched successfully. When then use that stitched image as the label or ground truth when building our training dataset for the supervised portion of our training. We refer to the image generated by our stitching model as $\hat{y}\in R^{H\times W\times 3}$.

Generally, stitching is an operation that takes a set of images and produces a seamless output panoramic image of the scene. Ideally, the output image should be able to preserve the content of each of the original images individually.

We define the parametric function F that maps a set of input images to its target $T^{i}$ [Equation 1]. The central idea of our method is to find the optimal parameter θ of F through an iterative learning process. Let $X^{i}={\left\{X^{i,k}\right\}}_{k=1}^{K}$ be a set of images captured simultaneously from *K* cameras at a given moment *i*. Here, $X^{i}$ refers to the set of warped images and we the notation shown below for clarity.
(3)$$F\left(X^{i},\theta \right)\to T^{i}$$

There are two ways to optimize the parameter θ through deep learning, namely, supervised and unsupervised learning. Because deep learning-based stitching (deep stitching) is a relatively young field, there are not many publicly available datasets. Although there are a few, we found none that met our requirements with regard to the number of cameras and lack of constraints on camera position. Rather than create an exhaustive dataset, we undertook the challenge of constructing a method to work with limited amounts of data. It is for this reason that we combined both methods of learning for parameter optimization, creating what is known as a semi-supervised network. We explain the process used to compute the target for our training set in more detail in the following section.

#### 3.2.2. Objective Function for Supervised Learning

Initially, we use supervised learning with known target images, with the primary goal being the reproduction of the target image. To create the ground truth data, we use multi-band blending with 20 bands to obtain high-quality mosaics through a slow traditional process. Thus, for supervised training, F from Equation (Equation 3) is a 20-band multi-band blending of all images. Using the result of this blending, we then compute the loss of our network using the supervised loss function ($L_{sup}$) defined below:(4)$$\begin{array}{c} L_{sup}\left(y,\hat{y}\right)=\lambda_{c}L_{c}\left(y,\hat{y}\right)+\lambda_{tv}L_{tv}\left(\hat{y}\right)+\\ +\lambda_{r}L_{r}\left(y,\hat{y}\right)+\lambda_{p}L_{p}\left(y,\hat{y}\right) \end{array}$$

$L_{c}\left(.,.\right)$ is the content loss function as defined in Equation (Equation 6).$L_{p}\left(.,.\right)$ is the mean square error per pixel loss function as defined in Equation (Equation 9).$L_{tv}\left(.,.\right)$ represents the total variation as defined in Equation (Equation 8)$L_{r}\left(.,.\right)$ is the reconstruction loss function as defined in Equation (Equation 7)$\lambda_{c}$, $\lambda_{tv}$, $\lambda_{r}$, and $\lambda_{p}$ control the contribution of each function to the overall loss calculated by the model.

#### 3.2.3. Objective Function for Unsupervised Learning

For the unsupervised training, we aim to optimize the model based on the content loss and the total variation. We first compute the feature map of the warped images for each camera, find the mask for the region in the warped image that contains the image data of the corresponding camera, then use that mask to select the region of interest from the generated output for comparison. The unsupervised loss function ($L_{u}\left(.,.\right)$) is defined as follows:(5)$$\begin{array}{c} L_{u}\left(\hat{y},\omega \left(X\right)\right)=\sum_{i=0}^{k}\lambda_{c}L_{c}\left(\hat{y}⊙m_{i},\omega_{i}\left(X_{i}\right)\right)\\ +\lambda_{tv}L_{tv}\left(\hat{y}⊙m_{i}\right) \end{array}$$
where $L_{c}$ is the content loss function that allows the error between the feature map extracted from the warped images $\omega_{i}\left(X_{i}\right)$ and generated image $\hat{y}$ to be minimized and $L_{tv}\left(.\right)$ computes the total variation of the generated output image [41,42]. The mask $m_{i}$ corresponds to the position of the non-zero pixels for each camera, ⊙ is a pixel-wise operator, and $\lambda_{c}$ and $\lambda_{tv}$ are the weighted coefficients for the content loss and total variation loss, respectively.

#### 3.2.4. Formulation of Each Objective Function

Content Loss Function ($L_{c}$) is widely adopted in the literature [41,42] to measure the similarity of the high-level features between the target *y* and output $\hat{y}$. The features are extracted using a pretrained network Φ such as VGG on ImageNet, with any trailing densely, fully connected, or perceptron layers removed to expose the final feature map of the convolutional layers. It is defined as follows:(6)$$\begin{array}{c} L_{c}\left(y,\hat{y}\right)=E\left[{∥\Phi \left(y\right)-\Phi \left(\hat{y}\right)∥}_{2}^{2}\right]. \end{array}$$

Reconstruction Loss ($L_{r}$) is used for supervised training to ensure the consistency of data between the gradients of both images. Similar to the per-pixel loss, the reconstruction loss computes the mean squared error of the gradient of both images. It is more robust than the naive MSE, and ensures that local pixel variations remain consistent between the target and the generated image. The ∇(.) operator indicates the gradient in Equation (Equation 7).
(7)$$\begin{array}{c} L_{r}\left(y,\hat{y}\right)=E\left[\sum_{i,j}{∥\nabla y_{i,j}-\nabla {\hat{y}}_{i,j}∥}_{2}^{2}\right] \end{array}$$

Total Variation ($L_{tv}$) is a regularizer used to ensure that the resulting output image is smooth.
(8)$$\begin{array}{c} L_{tv}\left(\hat{y}\right)=E\left[\sum_{i,j}{\left({\left({\hat{y}}_{i+1,j}-{\hat{y}}_{i,j}\right)}^{2}+{\left({\hat{y}}_{i,j+1}-{\hat{y}}_{i,j}\right)}^{2}\right)}^{\frac{\beta }{2}}\right] \end{array}$$

Per-pixel Loss Function ($L_{p}$) is the simple mean squared error between the two images. It ensures that the pixels of each image (as opposed to their features) are as similar as possible, and adds a level of refinement to the output.
(9)$$\begin{array}{c} L_{p}\left(y,\hat{y}\right)=E\left[{∥y-\hat{y}∥}_{2}^{2}\right] \end{array}$$

However, as the number of known target images is limited, it is necessary to switch to an unsupervised network in order to work with unlabeled data. Although there may not be proper ground truth labels for the majority of data, we can leverage the images provided as inputs to the model to make broad predictions as to the expected output. Specifically, our approach to the stitching problem at the unsupervised level is to view it as an image reconstruction problem in which each layer of the data represents a corrupt view of the scene, similar to [21]. The most relevant aspects here are the parallax-related errors, the camera exposure, and the image resolution. This implies that using per-pixel metrics such as the Mean Squared Error (MSE) and the Structural Similarity Index (SSIM), may not be ideal due to misalignment errors.

Instead, we turn to a customized loss function based on image perception, the inspiration for which is discussed in further detail in Section 3.2.2. This loss function uses a pretrained and frozen network following the InceptionNet (v3) architecture, and is trained for object recognition on ImageNet in the way discussed in [36]. We use this network to score the similarity between two images across a few different domains, including content and pixel value, as shown in Equation (Equation 5). Specifically, we take the warped image and use it to identify a mask to select the same pixel locations from the final output image. We then use the perception network to compare the warped image and masked output image in order to determine similarities in style and content, and take the amount of difference in each as a component of loss. As the images grow more similar in each category, the information from the warped image should be more present in the stitched one, which, if it is true for all warped images, leads to a final image that is a better representation of the true scene. The loss found from the differences can then be used to update the weights of the stitching model, though not the perception model.

#### 3.2.5. Generating the Dataset for Training

We use the global camera registration method to train our model to generate warped images, then compute the SandFall block and subdivide the tensor into patches of size 256×256 (in pixels) with a step size of 64 to allow overlapping between patches. Moreover, to enable the stitching model to capture global and local features during the training, we define multiple scale factors to resize the warping images. The scale factors allow a multi-level patch extraction to be performed on the final warping images, as illustrated in Figure 6. Scale factors are floating numbers $\alpha_{1..l}\in R$ defined such that the size of the output images is multiplied by $\frac{1}{\alpha_{i}}$.

## 4. Experimental Results

### 4.1. Dataset and Preprocessing

We used a large unlabeled dataset from a publicly available source by Walt Disney [10]. A few sample images from the dataset can be seen in Figure 7. This dataset was chosen because it is one of the few to utilize multiple (more than two) cameras in an unstructured manner. In addition, we created our own dataset using multiple synchronized cameras connected to Raspberry Pi 3Bs to record 1920×1080 pixel videos encoded with H264 in both indoor and outdoor scenes. We converted these videos into image sequences frame-by-frame. The cameras were the standard Raspberry Pi camera modules with Sony IMX219 8-megapixel sensors (Sony Corporation, Tokyo, Japan). The combined dataset contained a wide range of image resolutions, from low-resolution images with 256×256 pixels to high-resolution 5K images. The first step in our training process is data preparation, where we use a simple image registration process as proposed by Brown et al. [2] to warp input images into a common space. We then split that common plane into 256×256 subsets. use SandFall to compress them into a reduced array and prepare the appropriate masks.

### 4.2. SandFall Results

As a second preprocessing step, each set of input images $X^{i}$ is compressed and refined using our SandFall algorithm. We selected the user parameter to be n=5, which was determined experimentally. Figure 8 shows the contents of each layer after compressing images from fourteen cameras via SandFall. Specifically, it can be seen that in two cases there is no data loss at all (the fifth layer is empty), despite the storage array being reduced to just over one third the original size, from 42 channels to 15.

### 4.3. Experimental Setup and Training Parameters

Training and inference were conducted entirely via Nvidia RTX 3090 Graphical Processing Unit (GPU) and a Tensorflow 2.0 back-end for Keras and Python 3.6. The Central Processing Unit (CPU) for the system was an Intel 9700K at 3.6 Gigahertz (GHz) clock with 20 Gigabytes (GB) of Random Access Memory available, and the dataset was stored on a 512 GB 7200 rotations per minute (RPM) hard disk. The system parameters were as follows. As mentioned above, *n* was set to 5 for SandFall. We used the ‘Adam’ optimizer with default Keras settings and a $10^{-3}$ learning rate. The patch diameter chosen for the unsupervised loss was d=33, with a distance of 10 pixels per patch center. Finally, the model was trained for 100 epochs. These parameters were all determined experimentally. Each weight loss was set up experimentally during the training; we used $\lambda_{c}=0.1$, $\lambda_{p}$ = $10^{-5}$, $\lambda_{r}$ = 0.4, and $\lambda_{tv}$ = $4\times 10^{-7}$ for the content weight loss, the per-pixel loss, the reconstruction loss, and the total variation loss, respectively.

### 4.4. Model Performance

We evaluated the performance of our model against state-of-the-art methods for stitching inputs from unstructured camera arrays on a number of different image sets.

Table 1 compares the time required by our method to stitch camera images to two well-established stitching methods from the literature, namely, multi-band and seam-based stitching. As each of these approaches uses the same warped image set as inputs, it is possible to clearly evaluate the differences in stitching performance between them. From this experiment, it can be observed the number of cameras has little impact on the amount of time required by our system to generate an output. This is because SandFall fixes the input size regardless of the number of cameras. As a result, our method is much more scalable for systems with many cameras compared to other methods, which often need on the order of one second to stitch an image pair together even when using GPU acceleration [15,38]. It is important to note that we ran our method entirely on CPU in our experiments in order to more directly compare our approach to multi-band blending and seam-based stitching, which are recent approaches able to operate on unstructured camera arrays and are limited to CPU. Our results would likely be significantly faster if executed on GPU, as is the case for other neural networks. Even with this handicap, Our approach requires far less time to execute than the other methods tested, requiring less than one fifth the time of the method used to create our ground truth labels.

#### 4.4.1. Quantitative Comparisons

In order to compare our method to traditional stitching approaches, we compared the quantitative results of the stitching process. To do this, we quantified the artifacts in the stitched images using four performance metrics: the Learned Perceptual Image Patch Similarity (LPIPS) [33], Fréchet Inception Distance (FID) [35], Inception Score (IS) [34], and Improved Inception Score (SG) [36]. Here, we report quantitative results obtained by comparing the output of our model with those of multi-band and seam-based image stitching. We created a dataset that contains a variety of images with an emphasis on the number of cameras. The results of this analysis can be found in Table 2. Specifically, we evaluated performance using a number of metrics, as introduced in Section 2.5. The reasoning behind the selection of each of these metrics and what they convey can be found in Section 5.1. While our model does not outperform traditional methods in every case, it does on FID, and produces similar results to the existing methods in the others while executing far more quickly.

#### 4.4.2. Qualitative Comparisons

Here, we provide example outputs from each methodology for comparison in Figure 9. Our method results in outcomes with fewer or less noticeable artifacts than the others in most cases, although we admit this is more of a subjective comparison. Nonetheless, a number of common issues that traditional methods struggle with are solved through our technique, as shown in Figure 10. In particular, our approach can better address the problem of ‘broken objects’, where features are not properly aligned after being stitched together.

## 5. Discussion

### 5.1. Image Quality Assessment (IQA)

Except for the Inception Score, metrics used for image quality assessment are often designed around the availability of ground truth data for use as a reference or target *y* with the generated output $\hat{y}$. However, it is often difficult to obtain a ground truth in the context of an image stitching model for unorganized arrays of cameras. To mitigate this issue, we propose an extension of existing metrics that utilizes data that must be available, namely, the image input *X*.

The goal of many types of image generation is to create an image that contains the desired structure and information. When ground truth data are available, it is easy to compare the generated image to the ground truth using various metrics to score the similarity of content. However, as mentioned previously, in image stitching there is often no available ground truth. The content and structure of the final image should reflect all the information in the constituent images. With this in mind, we assume that an image is likely to be well stitched if the high-level features of the individual warped images can be drawn from the features space of that image; put another way, it should be possible to regenerate the input images from the stitched image. Therefore, our metrics for quality assessment measure the similarity between the stitched image and the individual warped images used as ground truth in the feature space. This is the primary reason that we have choose to use feature-based metrics such as the LPIPS [33], Inception Score [34], and FID [35], which are intended for these exact scenarios, instead of traditional metrics such as MSE, PNSR, and SSIM, despite the shortcomings of the former [33]. While pixel-based similarity metrics help to ensure that images are clear and sharp, their lack of correct content and structure means that they are ultimatly of limited value. However, as clarity and sharpness are important in images, we do include MSE in the loss.

Given a stitched image $\hat{y}$, warped images $\omega \left(X\right)={\left\{\omega_{i}\left(X_{i}\right)\right\}}_{i=1}^{k}$, and metric function $M_{p}$, we compute the final metric between the images by the taking the average:(10)$$\begin{array}{c} M\left(\hat{y},\omega \left(X\right)\right)=\frac{1}{k}\sum_{i=0}^{k}M_{p}\left(\hat{y}⊙m_{i},\omega_{i}\left(X_{i}\right)\right). \end{array}$$

We used the method described in Equation (Equation 10) to compare and evaluate the performance of our method against other stitching algorithms such as multi-band image stitching.

### 5.2. SandFall

One of the main issues in image stitching is time complexity, with real-time solutions being relatively rare [16,20]. While stitching a low number of images together in real time has already been accomplished under certain constraints [20], doing so for a large number has yet to be addressed. One of the major challenges is that pairwise stitching does not scale well with additional images, at best being $O(N*log_{2}\left(N\right))$ complexity [10], which, due to the high computational complexity of stitching even two images, makes this approach undesirable. Thus a method to ensure flat time when stitching multiple images is essential.

At the same time, many image stitching approaches, both traditional ones and those based on deep learning, have hard limitations on the number of images that can be processed simultaneously, at the very least requiring a known number. This obviously presents a challenge to unstructured camera arrays, which can have an unknown or differing number of cameras.

We have introduced SandFall to resolve both of the above issues simultaneously. By compressing multiple images into a fixed size, SandFall not only ensures that stitching takes the same amount of time regardless of the number of cameras, it allows for a set number of inputs that the network can expect. It does this with minimal loss of data, as shown in Figure 8; however, as the number of overlapping views in the scene increases, there is likely be at least some data loss. By ensuring that a minimum number of views are present prior to any data loss, however, SandFall mitigates the effects of this loss. This is not to say that data loss is nonexistent or unimportant; as SandFall is a naive implementation and its current implementation has no criteria for what values are selected or would be the best to preserve, we leave this question to future work.

#### Intrinsic Artifact Mitigation

Additionally, there is one more benefit over traditional methods when using our neural network for image stitching, namely, the removal of intrinsic artifacts such as noise and image pixel corruption from input images. Removing these artifacts often requires additional processing steps for multi-band and seam-based techniques, which adds to computational overhead. Using our neural network, the model can remove or mitigate these artifacts without requiring these extra steps; a comparison between methods based on the type of artifacts is shown in Figure 11. Specifically, it can be seen that our method is able to more easily recover from JPEG artifacts and noise than MBB and SBB and that it performs similarly with low-resolution inputs, all while requiring less time than either method.

## 6. Conclusions

In this work, we propose an image-stitching framework that supports an unlimited number of unstructured cameras, unlike other deep-learning stitching networks that only operate pairwise. As a result, our network throughput scales better with additional cameras than many existing state-of-the-art approaches. This improvement is thanks to our proposed SandFall algorithm, which enables accurate representation of perspectives from an unlimited number of cameras. To better handle parallax-induced misalignment errors, we leverage semi-supervised learning and a custom loss function to simultaneously handle both color variation and potential image artifacts. Finally, we compare our model’s output with the multi-band blending and seam-based approaches, finding that our method has similar performance and sometimes outperforms traditional algorithms in terms of quality and metrics. Thus, while there may be differences in the outputs between our method and the reference outcomes, they are generally very similar.

## Figures and Tables

**Figure 1 sensors-23-09481-f001:**
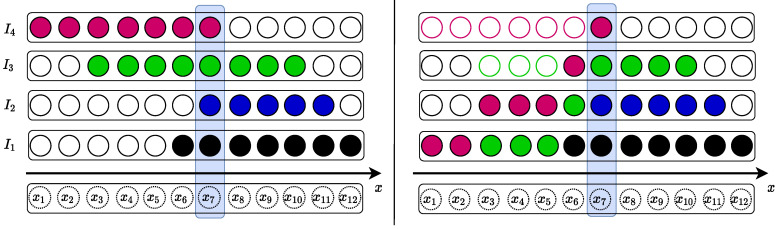
Illustration of 1D data representation of the SandFall method. The left side represents four pre-aligned images $I_{1-4}$ on the *x* axis, which normally requires a matrix size of (4×12) to preserve all of the information necessary for reconstruction with the naive representation. On the right side, we apply our SandFall data representation approach and reduce the matrix size to (3×12) while mitigating information loss.

**Figure 2 sensors-23-09481-f002:**
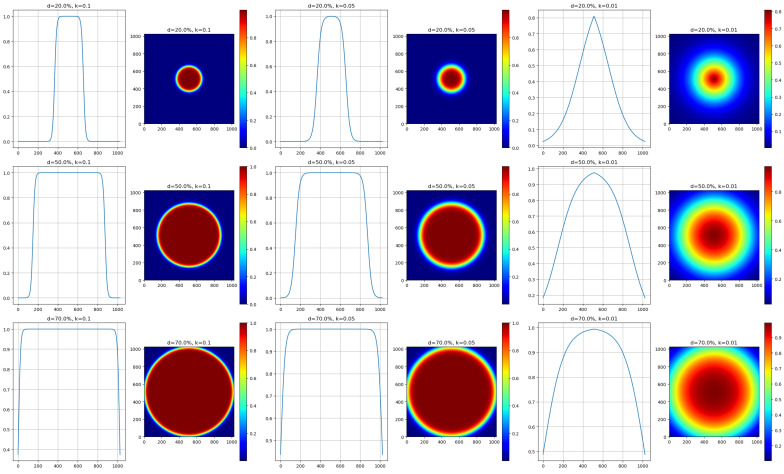
Illustrated variations of the inverted sigmoid function by different *k* values and *d* percentages, defining the response rate and relative width against the image diagonal. The first column plots the sigmoid curves with changing *d* values, while the second and third columns show the associated two-dimensional weighted masks. Rows represent *d* values at 20%, 50%, and 70% of the diagonal, alongside *k* values of 0.1, 0.05, and 0.01, demonstrating their influence on weight distribution in our SandFall algorithm.

**Figure 3 sensors-23-09481-f003:**
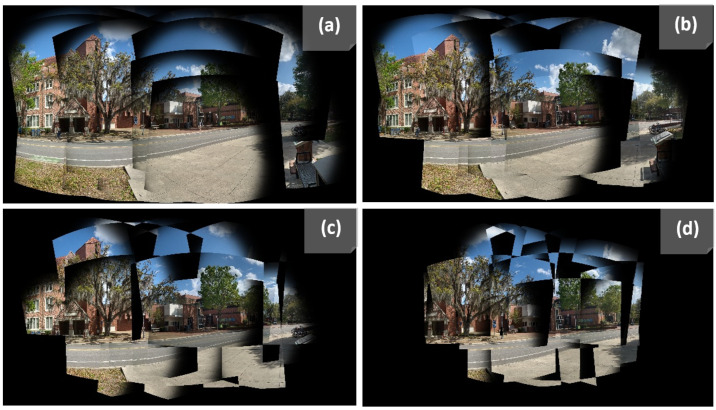
Progressive visualization of the SandFall method across four levels (**a**–**d**).

**Figure 4 sensors-23-09481-f004:**
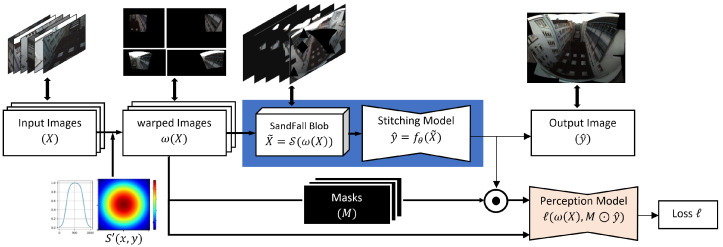
Overview of our proposed method. A set of images *X* captured simultaneously from several cameras is warped into a shared space ω(X). The SandFall method is applied to the warped images to ensure a fixed-size input. This input is then passed to our proposed stitching model to generate an output $\hat{y}$. A pretrained model was used to compare the perceptual similarity between the generated image and the target.

**Figure 5 sensors-23-09481-f005:**
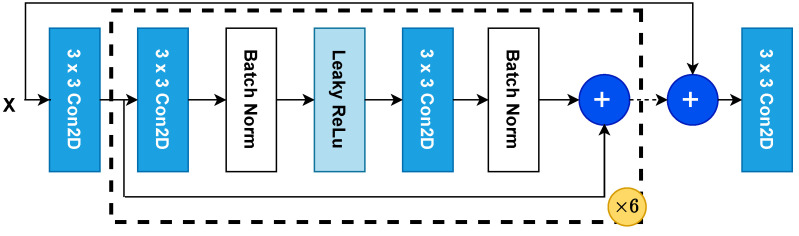
The overall architecture of our neural network.

**Figure 6 sensors-23-09481-f006:**
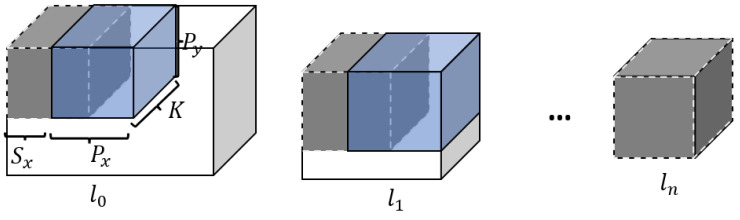
Illustration of the process used to generate patches for the training dataset. The images of the tensor are resized based on the scale factor until the resulting image has the same size as the patch. $S_{x}$ is the step size, $\left(P_{x}\times P_{y}\times K\right)$ is the patch size, and *K* corresponds to the depth of the SandFall block.

**Figure 7 sensors-23-09481-f007:**
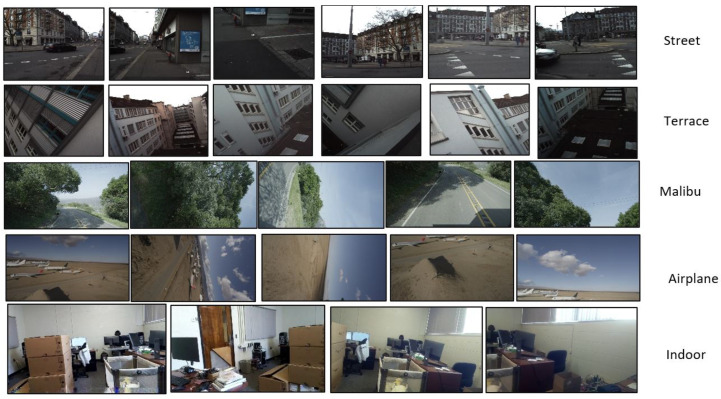
Sample dataset used for experimentation.

**Figure 8 sensors-23-09481-f008:**
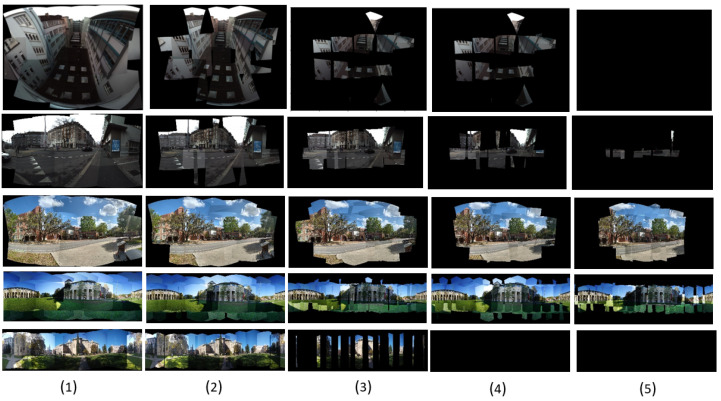
These images are the result of the data representation using the SandFall method. The original data consist of fourteen images. Using our method, the input sample is represented using five layers without any major loss. The fifth image represents the data of the fifth layer, which in this instance can be discarded without any information loss.

**Figure 9 sensors-23-09481-f009:**
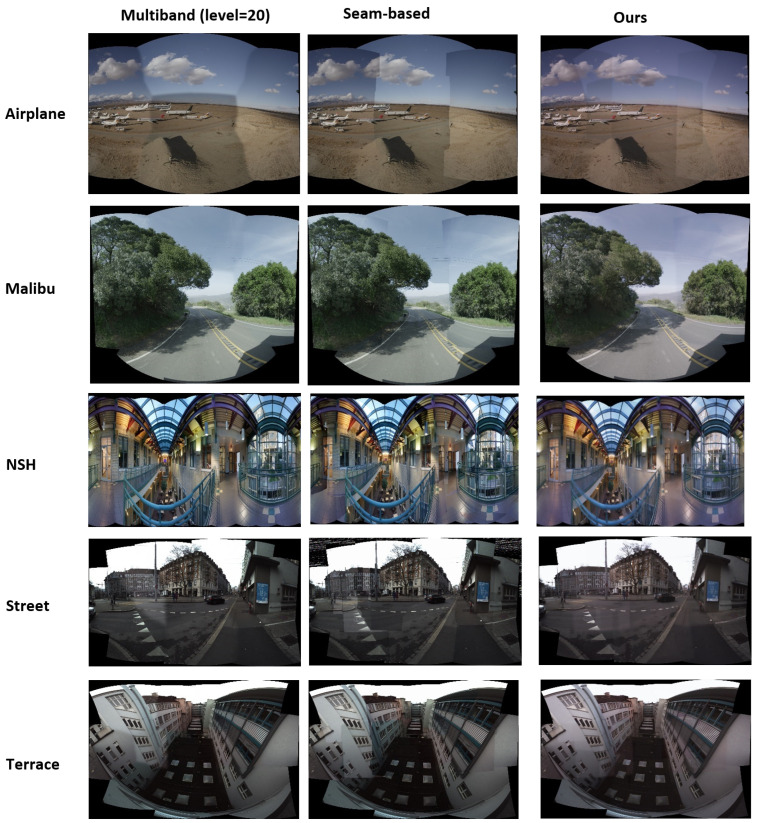
Comparative display of image stitching outcomes using three different methodologies. The columns represent the results from the Multiband (level-20), Seam-based, and Our proposed methods, respectively. Each row showcases the performance of these methods on a variety of datasets, including Airplane, Malibu, NSH, Street, and Terrace scenes.

**Figure 10 sensors-23-09481-f010:**
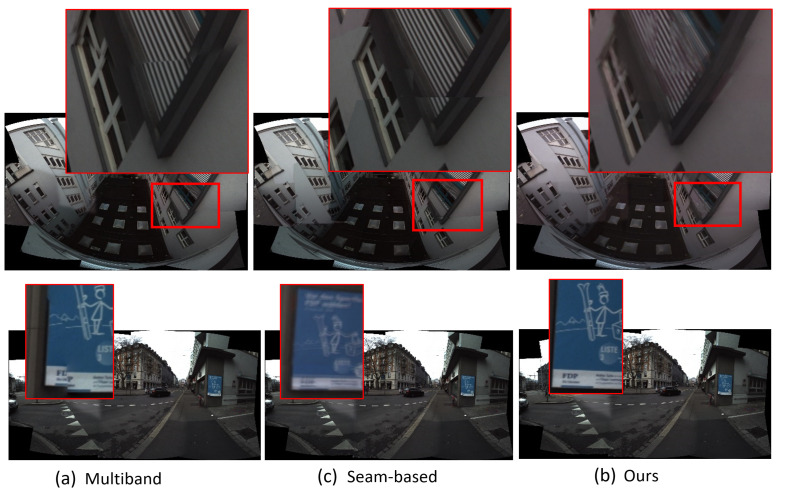
Comparative visualization of stitching quality across different methods. Subfigures (**a**), (**b**), and (**c**) display results using Multiband, Seam-based, and Our proposed methods, respectively. Each image is marked with red frames to emphasize regions of interest where stitching artifacts are most prevalent. These regions highlight the strengths and weaknesses of each method, particularly in maintaining structural integrity and seamless blending.

**Figure 11 sensors-23-09481-f011:**
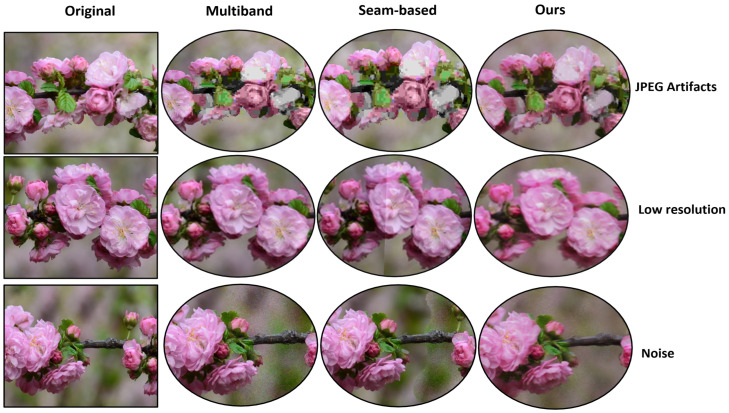
Experimental results evaluating the robustness of each method in the presence of artifacts. Experiments were conducted by manually introducing artifacts into the images prior to stitching. The first row simulates compression artifacts (JPEG), the second row simulates the stitching of low-resolution images into high-resolution ones, and the last row simulates the introduction of Gaussian noise in one of the images.

**Table 1 sensors-23-09481-t001:** Time required for Seam Based Stitching (SBS) and Multi-Band Blending (MBB) with 5 or 20 bands compared to our method.

Image Set	# of Cameras	Multi-Band Blending (5)	Multi-Band Blending (20)	Seam-Based Stitching	Our Method
Terrace	14	17.35 s	74.7 s	187.6 s	**12.77 s**
Malibu	5	24.8 s	101.19 s	126.4 s	**15.5 s**
Airplane	14	25.55 s	106.16 s	122.3 s	**15.14 s**
Street	5	16.89 s	81.6 s	177.6 s	**9.2 s**
Indoor	14	1.21 s	5.04 s	19.25 s	**1.80 s**
Averages	N\A	17.16 s	73.4 s	126.63 s	**10.89 s**

**Table 2 sensors-23-09481-t002:** This table compares the performance metrics of three stitching methods on five datasets. The performance metrics are computed as defined in Equation (Equation 10). LPIPS and FID are distance metrics, meaning that lower values are preferable, while higher values are desirable for IS and SG. Note that in this example multiband denotes 20-band Multi-Band Blending. The down arrow in front of the metric name (↓) means lower is better, while the up arrow (↑) means higher value is better.

	LPIPS ↓			FID ↓			IS ↑			SG ↑		
**Dataset**	**Seam**	**Multiband**	**Ours**	**Seam**	**Multiband**	**Ours**	**Seam**	**Multiband**	**Ours**	**Seam**	**Multiband**	**Ours**
Windmills	1.241	**1.172**	1.239	16.18	**15.45**	16.90	**4.10**	1.89	1.87	**1.41**	0.63	0.62
Terrace	0.531	**0.517**	0.533	5.66	5.60	**4.92**	**1.357**	1.337	1.344	**0.305**	0.290	0.295
Airplanes	1.238	**1.163**	1.234	11.74	**11.60**	12.32	**1.61**	1.57	1.48	**0.48**	0.45	0.39
Street	0.694	**0.67**	0.696	13.35	13.46	**11.67**	**5.84**	2.86	2.45	**1.76**	1.05	0.89
Malibu	1.177	**1.10**	1.174	16.48	16.91	**16.07**	**4.65**	2.57	2.83	**1.53**	0.94	1.04
**Averages**	0.9762	**0.9244**	0.9752	12.682	12.604	**12.376**	**3.5114**	2.0454	1.9948	**1.097**	0.672	0.647

## Data Availability

The data presented in this study are available on request from the corresponding author.

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
