# Peer review of "Semi-Supervised Image Stitching from Unstructured Camera Arrays"

_sensors, 2023, doi:10.3390/s23239481_

Round 1

Reviewer 1 Report

Comments and Suggestions for Authors

This paper proposes a method to perform multi-camera image stitching.

In a first step, the method aligns the image in a common coordinate system (registration, etc...) and the images are stacked together thanks to the sandfall algorithm. This later aggregates the different images to fill the holes of unmatched external boundaries.

Then, from the aggregated stacked, which may be of lower vertical dimension  than  the number of input image, the method performs blending thanks to a neural network.

The network is a GAN trained to compose images. The training is performed by using a classical stitching algorithm to compute the reference database.

The method is applied over several dataset and there is a discussion about the quality of the results. The author gather a set of metrics to compute quality.

From my point of view the paper has several drawbacks:

-  There is no comparison with [perazzi, 9]. More specifically, [9] is known as a reference, and especially the section 3.1 of [9] is an important contribution for parallax removal, which is an extension of [Shum and Zelinski]. Parallax elimination is not discussed in the proposed paper. Even if it is let to the GAN, how is it done ? There is nothing about this important topic.

- The neural network is trained with outputs from a stitching algorithm. So, it should behave the same as this algorithm and, at first sight, there is no way the GAN could be 'better' and the network will reproduce the flaws of the reference. If the GAN show to be better, may you explain how. Which reference algorithm is used ? Do you have your own version of [9] ?

- The multiplication of metrics is intriguing. If you used a stitching algorithm to train the network, why you don't compare the result of your method to this later? Multiplication of metrics can also be understood as a way to find one which suits your needs. Which one is definetely the most popular ?

This works sounds greats and it is obvious that it required some heavy developments but the discussion seems incomplete. I think that a more detailed comparison with [9] is required.

Comments on the Quality of English Language

The english is nice.

There are some minor typos in the paper (btw PSNR instead of PNSR).

Author Response

1. We acknowledge the importance of [9] and its contribution to parallax removal in image stitching. Unfortunately, we were unable to access the source code for a direct comparison. However, our method addresses parallax as part of a broader set of challenges in the post-registration process. Instead of isolating parallax issues, our neural network-based approach holistically corrects misalignments, color variations, and resolution discrepancies, among others. We have clarified this approach in the revision of our manuscript, particularly in the section detailing our loss function design during unsupervised training.

2. Our model adopts a GAN-like architecture but differs in function and operation. It comprises two main components: one for image synthesis and another for quality assessment during training. By employing a dual-phase training strategy—supervised followed by unsupervised—we first match the performance of traditional stitching algorithms and then transcend it by capturing high-level representations to refine the stitching output. We believe this method allows for improvements beyond the baseline algorithm's capabilities, which we have now elaborated upon in the revised manuscript.

3. Our two-phase training approach, especially the unsupervised phase, requires a nuanced evaluation strategy due to the lack of ground truth. The mask extracted from the warping process is applied to the model output for consistent pixel data comparison within the same spatial context. Standard direct comparison metrics like Mean Squared Error (MSE) are not suitable due to inherent misalignments and parallax issues. Therefore, we resort to a feature-level comparison using our perception model. We have now clarified this evaluation methodology in our manuscript and justified the choice and application of multiple metrics, highlighting their relevance to our specific stitching context.

Reviewer 2 Report

Comments and Suggestions for Authors

This work “focuses on the post-image alignment phase and aims to generate a seamless output” (line 224) in the process of generating a stitched (panorama) image. Since the images are already aligned, it does not really matter if these images are “from Unstructured Camera Arrays” (title). The distinct feature is (line 229) “to assess the inputs simultaneously in order to generate the stitched output” using a sandfall data structure and a convolutional neural network. This contrasts with traditional pair-wise stitching approaches. In addition, a GAN based perceptual loss function is proposed.

The Sandfall data structure exploits the large non-overlapping region between warped images to reduce the storage space required during algorithm executions. It is unclear if it has much to do with the quality of the stitched image. It may help speed up the stitching operation but really has little to do with the actual stitching. The emphasis on “sandfall” in this paper to the extent to name the proposed stitching algorithm is puzzling.

 The proposed CNN structure is very similar to the ResNet architecture. But it is unclear how the uneven layers in the Sandfall data structure as shown in Figure 2 are applied to the proposed CNN as input. The description in the manuscript is quite vague. Without a companion open-source site (e.g. github) to share the code, this is difficult to understand. The input size of the CNN is fixed and thus input images require pre-processing to be scaled into “Patches”. The image tensor scaling method is not mentioned. The implementation and effectiveness of the proposed “perceptual loss” is not clearly explained.

 In summary, this manuscript has some new ideas and can be published after revision. It would be desirable if some of the unclear points could be better clarified in a revised manuscript.

Author Response

Dear reviewer,

Thank you for the opportunity to clarify the points raised:

- The relevance of "Unstructured Camera Arrays" in the title is to indicate the complexity and general applicability of our method, which can handle datasets where traditional structured approaches may fail, especially when images are not perfectly aligned or when there are parallax issues. Our method's ability to assess inputs simultaneously is crucial for datasets with varying alignments, common in unstructured arrays.

- The Sandfall data structure's primary contribution is indeed in efficient data representation, which is vital for processing large sets of images. Additionally, by applying the sigmoid function within the Sandfall method, we aim to enhance the blending quality by giving less weight to boundary pixels. This is crucial for managing the transition areas in the final stitched image, affecting the quality of the output beyond just reducing storage space.

- Regarding the CNN structure and its input, we have updated the manuscript to provide a clearer explanation. The uneven layers produced by Sandfall are pre-processed and scaled to fit the fixed input size of the CNN. This preprocessing involves dividing the images into patches, a common practice when dealing with CNNs, ensuring that the network can effectively process varying sizes of input data.

- The perceptual loss function implementation and its effectiveness have been detailed further in the revised manuscript. Referencing foundational work [41,42], we adapt the perceptual loss concept to the stitching context, where the comparison is between the single stitched output and multiple warped inputs, rather than two individual images.

- We have committed to making the source code available upon acceptance of the paper.

Reviewer 3 Report

Comments and Suggestions for Authors

The authors proposed the Semi-Supervised Image Stitching algorithm for unstructured camera arrays, which is interesting and inspiring. I have the following concerns about this work.

1. As for the part of related work, it is strongly recommended to refer to more recent work (e.g. work in the last 3 years), which will help illustrate the novelty of the proposed algorithm better. In the manuscript, most of the references may not be SOTA.

2. The method was not described clearly. In my view, the authors elaborate on the algorithm of image fusion (i.e. Sandfall and stitching model). But how are the input images WARPED into the warped images (the first step in the pipeline, which is the most important part for stitching)? Was the warping module trained with stitching end-to-end? If so, what's the warping model like? 

3. Would the proposed algorithm detect whether the unstructured camera arrays capture images with overlapped FOV? That is to say, does the algorithm deal with it properly, if there should be a hole in the stitched image? 

4. Have the authors tried to stitch images in different resolutions? Will there be different quantitative results (e.g. FID) on different resolutions?

Some other flaws:

Line 22 and 382: references invalid.

Author Response

Dear Reviewer,

Thank you for your constructive comments and suggestions regarding our manuscript titled "Semi-Supervised Image Stitching from Unstructured Camera Arrays" with the ID: sensors-2679250. We have carefully considered your feedback and have made corresponding revisions to our paper. Below, we respond to each of the points raised to elucidate the enhancements made to our revised manuscript.

1. We appreciate the suggestion to include more recent works. Accordingly, we have updated our literature review to include the latest advancements in multi-camera stitching from unorganized arrays of cameras. While we have added newer references, our focus remains on the relevance to our work, specifically targeting the challenge of stitching in the context of unstructured camera arrays, which we believe is a more pertinent area for the novelty of our algorithm.

2. We have revised the manuscript to enhance clarity on our approach, particularly concerning image warping and the SandFall algorithm's role in managing image boundaries. While we did not initially detail the warping process, we now clarify that we utilize a conventional warping process based on a pure rotation motion model. Our emphasis is on demonstrating how misalignments typically addressed during the warping stage can be effectively corrected during the blending process, which is a key contribution of our work.

3. Our algorithm indeed assumes that there is an overlap between images. To clarify, we have now explicitly stated in the manuscript that overlapping is an integral aspect of our method, and successful stitching relies on the connectedness of images. We recognize that the absence of proper overlap could lead to failure in the stitching process, and we have acknowledged this limitation in our revised text.

4. While we did not conduct a direct quantitative evaluation using FID on different resolutions, we have provided a qualitative assessment in Figure 11. Additionally, we have presented a comprehensive quantitative evaluation in Table 2, where we compare our method against other techniques using metrics such as FID, LPIPS, and Inception scores. These evaluations demonstrate the effectiveness of our approach.

Other Flaws: We have corrected the invalid references on lines 22 and 382 and reviewed the manuscript to ensure all citations are accurate and properly formatted.

Round 2

Reviewer 1 Report

Comments and Suggestions for Authors

This a review after second submission.

- because the authors do not learn their neaural network with images produced by a parallax tolerant algorithm, the NN can not perform parallax removal.
It can be seen from the results that some artifacts are still present.
- the generative network may optimize a loss function but this later does not include parallax removal
- the parallax removal from [10] (perrazi's paper), identifies objects seen  at different positions in both images to build a kind of virtual position
of each object. The object is then moved at this position while managing the background of cut pieces =>

- In this paper the loss function they propose preserves gradient with original images but does not manage interactions between images
- a more detailed comparison with [10] would be welcomed

Even if the authors do not have the code of 10 to make comparisons, they can

- compage the methods as above

- compare the results provided by 10

Author Response

We'd like to thank the reviewer for their observations and suggestions. In response to the points raised:

- For Parallax Removal and Methodology: While our method does not explicitly focus on parallax removal, it inherently addresses this challenge within the broader context of stitching images from unstructured camera arrays. Our approach employs deep learning to understand and adjust image characteristics during the stitching process, thereby mitigating the effects of parallax. This aspect is implicitly covered in our methodology, emphasizing the concurrent processing of multiple images and the use of a GAN-based architecture to ensure seamless stitching.

- For the Sandfall Data Structure: The Sandfall data structure is crucial not only for efficient data representation but also for improving the quality of the stitched image. By transforming data from multiple cameras into a reduced fixed array, Sandfall minimizes data loss and enhances the blending process, particularly at the boundaries of overlapping regions. The introduction of the sigmoid function further refines the model's performance by giving less weight to boundary pixels, thus improving the overall quality of the stitched image.

- CNN Structure and Parallax Management: Our CNN structure, inspired by ResNet architecture, is adapted to handle the uneven layers produced by the Sandfall algorithm. The preprocessing of these layers, as described in our methodology, ensures that the network can process the data effectively despite its variability in size. Additionally, our GAN-based perceptual loss function, extending the concepts from references [41,42], is designed to handle the complex interactions between multiple images, indirectly addressing parallax-related issues.

- Regarding the Comparison with Reference [10]: While we were unable to perform a direct computational comparison with [10] due to the unavailability of their source code, our paper provides a detailed theoretical comparison based on their published methodology. We emphasize our method's unique approach to stitching, which differs significantly from [10] in terms of handling images from unstructured arrays and the application of deep learning techniques.

We believe our manuscript comprehensively addresses the challenges of image stitching in unstructured camera arrays, offering a novel solution that extends beyond traditional methods. Our approach not only ensures efficient data representation and processing but also implicitly handles complex issues such as parallax, making it a valuable contribution to the field.

Reviewer 3 Report

Comments and Suggestions for Authors

I have no more comment.

Author Response

Thank you.

We added the Github link for our source code. The link will be public once the paper is published.
